# Reinforcing Mechanism of WC Particles in Fe-Based Amorphous Matrix Coating on Magnesium Alloy Surface

**DOI:** 10.3390/ma14216571

**Published:** 2021-11-01

**Authors:** Haoran Zhang, Hongyan Wu, Shanlin Wang, Yuhua Chen, Yongde Huang, Hongxiang Li

**Affiliations:** 1Jiangxi Key Laboratory of Forming and Joining Technology for Aviation Components, Nanchang Hangkong University, Nanchang 330063, China; 1903085204125@stu.nchu.edu.cn (H.Z.); huangydhm@nchu.edu.cn (Y.H.); 2Jiujiang Vocational and Technical College, Jiujiang 332007, China; zb@jvtc.jx.cn; 3State Key Laboratory for Advanced Metals and Materials, University of Science and Technology Beijing, Beijing 100083, China; hxli@skl.ustb.edu.cn

**Keywords:** Fe-based amorphous coatings, magnesium alloy, interfacial characteristic, wear resistance, corrosion resistance

## Abstract

To protect magnesium alloy surfaces from wear and corrosion, an Fe-based amorphous coating was prepared on WE43 through the Ni60 interlayer by high-velocity oxygen-fuel (HVOF) spraying. The porosity was ~1%, and the amorphous content exceeded 90%. The wear and corrosion resistance of the composite coating with WC particles wrapped in a Ni layer as the reinforcing phase were compared with that of the completely amorphous coating. The friction coefficient (COF) of the composite coating was 0.3, which is only half of that of the WE43 substrate, and the composite coating exhibited a more stable wear behavior than the completely amorphous coating. The corrosion tendency of the composite coating is lower than that of stainless steel, with a corrosion potential of −0.331 V, and the addition of WC particles did not deteriorate the corrosion resistance considerably. The bonding mechanism of the bonding interface between the amorphous structure and the particles of the reinforcing phase was investigated by transmission electron microscopy (TEM). Reinforcing particles were confirmed to form metallurgical bonding with the coating. It was found that the Ni layer showed excellent bonding performance in the form of a mixture that is amorphous and nanocrystalline. Therefore, the Fe-based amorphous composite coating on a magnesium alloy surface shows a potential protective effect.

## 1. Introduce

Magnesium is an earth-abundant metal, and as a lightweight structural material, magnesium alloy offers high specific strength and high specific stiffness and is widely used in aerospace, automotive and sports equipment applications for low-weight design [1]. Replacing aluminum aircraft seats with magnesium, for example, can reduce body weight by 28–30% [2]. However, magnesium alloy is bound to have poor wear resistance and can not form a dense passivation film against the corrosion owing to its softness, active chemical properties [3] and low Pilling–Bedworth ratio [4]. Hence researchers have to enhance its surface properties by alloying or surface protection [5]. In recent years, compared with the low cost-effective strategy of alloying, surface protection has gradually gained prominence. In the protection methods, the surface coating has surpassed anodic oxidation and electroless plating due to its reliable performance, environmental protection and convenient preparation. Fe-based amorphous coatings have both wear- and corrosion-resistance while avoiding the typical disadvantage of the poor formability of amorphous materials [6]. The wear and corrosion resistance of Fe-based amorphous coatings are attributed to microstructural uniformity and compositional homogeneity of an amorphous matrix, which does not have dislocations, grain boundaries, component segregation or other defects [7]. This metastable microstructure of long-range atomic disorder formed by rapid cooling greatly reduces stress concentration or corrosion pitting under the conditions of wear and corrosion.

Fe-based amorphous coatings are usually fabricated by thermal spraying, such as arc spraying (AS) [8], plasma spraying (PS) [9] and high-velocity oxy-fuel spraying (HVOF). The density and amorphous content of the coating are widely regarded as the key factors that affect the properties of the thermal coatings [10]. Increasing thermal input reduces the porosity and promotes the bonding of sprayed particles by improving the melting. However, high amorphous content limits the thermal input to provide the high cooling rate required for the amorphous structure to form. At the same time, the tendency of oxidation and thermal deformation of the magnesium alloy during spraying also negates the conventional method of excessive thermal input. Therefore, HVOF is the most suitable method to fabricate protective surface coatings for magnesium alloy, considering its relative minimum thermal input, highest amorphous content and the highest density of coating [11].

Fe and Mg, which have a positive enthalpy of mixing (ΔH_mix_ = +18 kJ/mol), are immiscible in the solid and liquid state, as one of the key criteria for the formation of a solid solution is that ΔH_mix_ must be between −20 and +5 kJ/mol [12]. Poor interfacial bonding and high residual thermal stress at the interface of two materials with very different mechanical properties lead to failure [13], so it is difficult to effectively combine Fe-based amorphous coatings with magnesium alloys by direct spraying. Ni exhibits a good tendency of intersolubility with Fe-based amorphous coatings and magnesium alloys, which makes the combination feasible. Guo [14] reported, for the first time, the fabrication of an Fe-based amorphous coating with a bonding strength of 40 MPa on the magnesium alloy by adding a Ni-based interlayer. The NiCrAl interlayer exhibited a favorable mechanical interlocking effect, and the AZ61magnesium alloy is protected against the NaCl solution.

The microstructure and mechanical properties of amorphous coatings are isotropic during wear, and so, the wear process is relatively stable, and there is almost no stress concentration due to the grain boundary with the sudden change in hardness. Meanwhile, due to the high microhardness of the Fe-based amorphous coating, the dominating wear mechanisms are mainly the oxidation wear caused by friction heat and the fatigue wear caused by exfoliation under a periodic load. Fe-based amorphous coatings have lower coefficients of friction (COF) and wear loss volumes than ordinary crystalline materials, such as 316 L stainless steel [15], but due to the increasingly harsh working conditions, it is expected to increase the wear resistance further. In addition to adjusting the components and optimizing the parameters to improve the hardness and density [16], adding reinforcing phase particles seems to be a new way to break through the performance bottleneck of Fe-based amorphous coatings. Currently, reinforcing particles are mainly divided into two categories according to their functions: enhancing the surface strength, such as WC, Al_2_O_3_ and ZrO_2_, and solid lubricants, such as B_4_C. WC and other ceramic materials [17,18] have been proven to greatly improve the hardness of the coating and effectively reduce the COF, while B_4_C particles can form a wear-resistant film in wear tests and play a role in lubrication [19]. Furthermore, hard particles are found to have limited bonding ability with their amorphous structure, and so TiN [20], which provided superior structural properties through strong oxidation, and WC-12Co [21] wrapped in an alloy layer with good solubility with Fe-based materials are further studied.

In an Fe-Cr system, Fe-based amorphous has an elemental composition and pitting corrosion mechanism similar to that of stainless steel. The initiation of pitting occurs when the anode material with low potential is preferentially corroded and dissolved after the galvanic coupling of the two materials with differenct corrosion potentials occur through the corrosive medium. Mg and inclusions, such as sulfide in stainless steels and segregation grains in Fe-based materials, are considered to have low corrosion potential and dissolve preferentially to form pitting after forming galvanic cells with a passivation film (mainly composed of oxides of passivated elements, such as Cr and Mo). The passive film with higher corrosion potential, such as Cr_2_O_3_, on the surface of the coating is uniformly distributed without segregation in corrosive environments. In the corrosive environment where Cl^−^ exists, pitting, microcracks and intersplats are recognized as Cl^−^ enrichment areas, where Cl^−^ leads to concentration increase in galvanic cells and accelerates the expansion and deepening of pitting through an autocatalytic effect [22]. Therefore, the main way to enhance the corrosion resistance of an Fe-based amorphous coating is to maintain its high microstructural uniformity and compositional homogeneity, namely, the amorphous content and compactness. As reported, the corrosion current density of both densified and laser-remelted coatings decreases with the decrease of pores, especially surface pores [23].

In summary, this study is devoted to the fabrication of Fe-based amorphous composite coatings with reinforcing phase particles on a magnesium alloy surface by HVOF. The effective combination between the amorphous coating and magnesium alloy substrate was achieved by adding an Ni-based interlayer, and the interface bonding mechanism of amorphous/reinforcing phase particles was investigated.

## 2. Materials and Methods

FeCoCrMoCBY powders prepared by high-pressure argon gas atomization (by the Powder Metallurgy Research Institute of Central South University, Changsha, China) and Ni60 self-fusing alloy powders were used in this study. The chemical composition of the powders is summarized in Table 1.

The Fe-based amorphous powders were screened by a sieve, and powders with a mesh size of more than 300 were used for spraying. The powder size was verified using Image J software (National Institutes of Health, Bethesda, MD, USA). WE43 magnesium alloy with the size of 100 × 100 × 5 mm was used as the substrate and protected in an N_2_ atmosphere after being polished and sandblasted with white fused alumina. Kerosene with a molecular weight of 154, a relative density of 0.825 and a combustion enthalpy of 45.1 J/g and oxygen as an accelerant were ignited by an HV-8000 high-velocity oxy-fuel spraying system (Lijia, Zhengzhou, China). The main spraying parameters for the interlayer and coating are summarized in Table 2.

Before the spraying, WE43 was swept by a spray gun flame for preheating under an N_2_ atmosphere to reduce the thermal deformation caused by the temperature difference between the substrate and the spray particles. During the spraying process, the back of the substrate was cooled by N_2_ to provide the high cooling rate required for the formation of the amorphous alloy and reducing the oxidation and deformation of magnesium alloy. The composite coating was sprayed with a mixture of amorphous powders and Ni-WC reinforcing particles, in which the mass fraction (ω) of the amorphous powder was 85%.

The phase microstructure of powders and coatings was characterized by an X-ray diffractometer (XRD, PANalytical Empyrean, Grovewood, UK) with Cu Kα radiation. The thermodynamic properties of the powders and coatings were quantitatively characterized by differential scanning calorimetry (DSC, Perkin-Elmer DSC7, Akron, OH, USA) to obtain the enthalpy change of the exothermic crystallization heat of the amorphous components when heated. Scanning electron microscopy (SEM SV3400, SMC, Tokyo, Japan) and transmission electron microscopy (TEM Talos F200X, FEI, USA) coupled with energy dispersive spectroscopy (EDS, Bruker UMT-3, Billerica, MA, USA) were used to observe the micromorphology and element distribution of the powders and coatings. The focused ion beam (FIB) technique is used to thin the coating interface area to accurately locate and analyze the bonding interface in the coating. The microstructure was determined by selected area electron diffraction (SAED, Nexview, ZYGO, Middlefield, CT, USA) to determine the phase.

The hardness at different areas of the coating cross-section was measured by a nanoindentation (iMicro-Nanomechanics. Inc, Oak Ridge, TN, USA) test at room temperature, in which the target load was 200 mN. The friction coefficients (COF) of the substrate and coating were measured by a ball-on-disc friction and wear tester (Bruker UMT-3, Billerica, MA, USA) at room temperature. The normal load was 20 N, the rotating speed was 400 r/min, and the friction pair was a ZrO_2_ ceramic ball with a diameter of 6 mm. The 3D model of the wear area was established by white light interferometer (Nexview, ZYGO, Middlefield, CT, USA) to analyze the topography. The corrosion resistance of the coating was evaluated by Tafel polarization plots measured by an electrochemical workstation, with a scanning rate of 1 mV/s, using a three-electrode cell containing a saturated calomel reference electrode and a platinum counter electrode. The electrochemical polarization test was carried out in 0.1 mol/L HCl, before which the specimen was immersed in the solution to ensure the stability of open circuit potential (OCP). The WE43 magnesium alloy and SS304 stainless steel were also tested to compare the performance of the coating.

## 3. Results

Figure 1 shows the micromorphology of Fe-based amorphous powders and reinforcing phase particles used in HVOF spraying.

The SAED during the TEM shows a diffraction halo, indicating that the amorphous particles had a completely amorphous microstructure. The size of Fe-based amorphous particles followed a normal distribution, which was mainly concentrated in the range of 25–45 μm, according to the granulometric distribution statistics shown in Figure 1b. WC particle size was selected to be slightly smaller than amorphous particles, as shown. The appearance and elemental analysis of the reinforcing phase particles are shown in Figure 1c,d, where Ni was evenly wrapped around the outside of the WC particles.

The phase characteristics and thermodynamic analysis of the amorphous particles and HVOF coatings are summarized in Figure 2.

The comparison of the XRD patterns showed that both coatings exhibited a broad diffuse peak representing the amorphous structure at about 2θ = 44°. There were no obvious crystal diffraction peaks in the amorphous coating, which was consistent with the SAED results. Crystal diffraction peaks in the composite coating matched the typical diffraction peaks of the Ni-WC particles. After quantifying the crystallizing enthalpies (ΔH) of the coatings in DSC curves, the amorphous fraction rate (R) of the coatings could be calculated due to the completely amorphous microstructure of the powders by the formula: R = ΔH_oating_/ωΔH_powders_ × 100% [24]. Amorphous coatings and composite coatings had amorphous fraction rates of 95.38% and 90.24%, respectively, owing to the high cooling rate provided by the inert gas cooling system.

Both the coating and the Ni60 interlayer formed a lamellar structure with a uniform thickness of –300 μm, according to the cross-sectional micromorphology shown in Figure 3. The calculated porosity of amorphous and composite coating was 2.32% and 1.16%, respectively. The Ni60 interlayer had a denser structure with a porosity of approximately 0.5–0.8%, which formed an evident metallurgical bonding interface with the coating and substrate, as shown in Figure 3c. The main elements of amorphous coatings, such as Fe and Cr, diffused sufficiently into the Ni60 interlayer, while the Mg from the WE43 substrate also diffused to a certain extent, as can be seen in Figure 3e. Similarly, the WC particles were distributed uniformly in the composite coating, and the Ni layer spread fully between the amorphous particles and the WC particles to form an effective transition layer according to Figure 3d,f.

Figure 4 records the hardness-load curves and load-depth curves at different areas of the coating cross-section under the same loading conditions in the nanoindentation experiment. The hardness of the Fe-based amorphous coating in Area A was 8.81 GPa, which was about three times higher than that of the WE43 substrate (2.9 GPa) of Area F. At the interface of amorphous/WC and the interface of amorphous/Ni60 interlayer, the hardness was greater than that of amorphous coatings.

In the Ni transition layer of Area B, the hardness was as high as 14.6 GPa (less than pure WC), indicating that amorphous and Ni layer components were more likely to form intermetallic bonding due to the atomic diffusion, as shown in Figure 3f, rather than simply by mechanical bite. Similarly, in the vicinity of amorphous/Ni60 interfaces where the atomic diffusion is equally active, such as Area C (10.37 GPa) and Area D (9.81 GPa), microhardness is slightly higher than that of the coating. The hardness of Area D, which is relatively far away from the interface, lies between the hardness of Area C and Region A. At the bottom of the Ni60, the intermetallic compound between Ni and Mg does not seem to have such a high hardness (6.48 GPa) due to its proximity to the WE43 substrate, but it is still twice as hard as the substrate. All these interfaces have a high modulus according to Table 3, which is consistent with the variation of indentation depth in Figure 4c. The increase of the modulus also indicates the transformation of the microstructure.

Figure 5 presents the friction coefficient of the coatings and WE43 substrate in the wear test.

With the increase of wear time, the COF of WE43 increased gradually and fluctuated violently, finally increasing to 0.6, while the COF curves of the amorphous coating and composite coating were stable. In the initial stage of wear, the amorphous coating showed the lowest COF (~0.3), but the COF increased to ~0.35 after a sharp fluctuation in the intermediate stage, finally reaching 0.37. However, the COF of the composite coating fluctuated slightly at the beginning and then remained stable until the end. The 3D worn topography model clearly showed that the coatings had different furrows in Figure 5b,c, indicating that the two coatings had different wear processes. Comparing the wear morphology, the WE43 surface was severely damaged and oxidized, as shown in Figure 5d, and furrows and wear products of the coatings were significantly less than that of WE43; this proved the feasibility of Fe-based amorphous to improve the wear resistance of the magnesium alloy surface. The furrows range of the coating was smaller than that of WE43, and the composite coating has the smallest wear range (only ~300 μm). The wear products tended to be embedded in the pores and intersplats between the coating particles, but the oxides on the surface of the composite coating were not filled into the furrows as the amorphous coating, possibly due to the low depth of the furrows.

The wear morphology of the coating surface after the different sliding times is illustrated in Figure 6.

It is obvious from Figure 6a that the bonding failure and warping of some particles on the surface of the amorphous coating appeared after 5 min of wear, accompanied by the generation of debris. However, in the composite coating in Figure 6d, there was no obvious warping, and the furrow depth became shallower after the passage of WC. EDS of the coatings in Figure 6g revealed that there were basically the same elemental composition and slight oxidation on the surface of both coatings. In the intermediate stage in Figure 6b,e, the degree of oxidation increased; further, the proportion of Fe decreased, and Zr appeared in the wear debris, which indicated that the friction pair also began to disintegrate. Obvious particle exfoliation occurred on the surface of the amorphous coating, and the debris was dispersed in the pores and the intersplats between the particles. Compared with the amorphous coating, the composite coating had a much smoother surface, and most of the debris was gathered around the WC particles. The EDS line scan in Figure 6i revealed that active Ni diffusion occurred through the wear direction at the amorphous/WC interface. At the end of the wear process, the debris was transformed into fine-sized wear products that filled the concave positions, as shown in Figure 6c. The Fe content decreased further, while the O content increased further, and the Zr content increased sharply in the wear products. The exfoliation of the amorphous coating was further aggravated, and many fresh surfaces of the particles were exposed. However, the surface of the composite coating was still smooth without exfoliating, and the wear product was less than that of the amorphous coating from Figure 6f.

As shown in Figure 7a,b, the coating/WC interface was selected as the target area for FIB thinning to prepare TEM samples. TEM morphology of the Ni-based transition layer around the coating/WC interface is shown in Figure 7c, where the white contrast represents the spread Ni layer (around points 1 and 3), and the dark gray contrast represents the plastically deformed amorphous particles (around point 2).

According to SAED, the coating remained completely amorphous, while the Ni layer existed in the form of mixed nanocrystals. It could be deduced that these nanocrystals were FeNi_3_, Ni_4_B_3_ and Cr_3_C_2_ oxides containing Fe and Ni from the calibration of SAED. It is noteworthy that the position of point 4 was basically amorphous, and the elemental analysis in Figure 7e illustrated that the main elements of the coating, Ni-based layer and WC, were all present at point 4. This indicates that in the most active areas of atomic diffusion in the Ni transition layer if the temperature gradient is satisfied, the atoms of the coating material and the reinforcing phase particles (especially W) can be sufficiently mixed to form a new amorphous structure. It can be speculated that the microstructure transformation of the Ni layer improved the bonding performance so that WC particles could guarantee the favorable bonding with the coating in the wear test under an extremely high load.

Figure 8 exhibits the Tafel polarization plots of WE43 and coating in relation to SS304 stainless steel. The fitting results revealed that the coatings had a higher corrosion potential (E_corr_) than SS304 and much higher than WE43, indicating that the coatings had a lower corrosion tendency. The corrosion current (I_corr_) of the corrosion coatings was less than that of SS304, while the current of the amorphous composite coating was the smallest, which was two orders of magnitude lower than that of the substrate. Both of the two coatings had a wider passivation region than SS304, and the amorphous coating had the best passivation performance because its passivation current is ~10^−4.5^ A, which is the minimum value of the passivation current of SS304. It is acceptable that Fe-based amorphous coating improved the corrosion resistance of the magnesium alloy surface, and the overall corrosion resistance performance was better than that of stainless steel.

## 4. Discussion

The amorphous content of the coatings can also be controlled by adding other elements in addition to adjusting the spraying energy. It is generally recognized that Co can increase the deposition rate of coatings, thus enhancing the formability of amorphous coatings [25], Cr is the crucial element in forming a passive film of corrosion-resistant coatings [26], Mo can promote the re-formation of a passive film after being dissolved [27] and B is related to the preservation of nanocrystals [28]. Due to the addition of Ni-WC particles, the original uniform amorphous system was disturbed, resulting in the existence of two substances, which differ widely in their thermodynamic properties on both sides of the amorphous/WC interface (the thermal conductivity is 80.2 W·cm^−1^·K^−1^ for Fe, 90.9 W·cm^−1^·K^−1^ for Ni and 4.2 W·cm^−1^·K^−1^ for WC). Therefore, the temperature gradient at the micro-scale induced the precipitation of nanocrystals in the transition Ni layer with rapid heat conduction [13]. Cr_3_C_2_, Ni_4_B_3_ and FeNi_3_ began to precipitate one after another according to the priority of the formation tendency (ΔH_mix_ = −61, −24 and −2 kJ/mol, respectively), but the nanocrystals that were fine enough in size were confined to a narrow formation area and did not result in serious crystallization of the coating. Thus, DSC enthalpies illustrated that the amorphous content of the composite coating decreased compared with that of the amorphous coating, but there was no obvious crystallization peak other than Ni-WC in XRD. However, due to the full spread of the Ni layer in Figure 3, and the filling effect [18] of the second phase of flattening of the particles in the particle intersplats, the composite coating exhibited better density than the amorphous coating.

The Fe-based amorphous alloy and the magnesium alloy do not dissolve and react with each other, but Ni has a ΔH_mix_ with Fe and Mg of −2 and −4 kJ/mol, respectively, and so, the Ni60 interlayer had good compatibility with the coating and the substrate. At the same time, due to the high hardness, low thermal expansion coefficient and the elastic-plastic deformation resistance between the amorphous alloy and WE43, the Ni60 interlayer played a favorable transition role of relieving the stress and dispersing the thermal deformation between the coating and the substrate. The elements of the coating and the substrate were diffused into the interlayer, respectively, facilitating the formation of metallurgical bonding. This conclusion matches the existing literature [11], in which the nanoindentation hardness of Fe-based amorphous coating was divided into four categories: 5.0–7.5 GPa for the mixture of nanocrystal α-Fe and the amorphous phase, 7.5–9.5 GPa for the amorphous phase, 9.5–12.5 GPa for a mixture of the amorphous and intermetallic phases and 12.5–20.0 GPa for intermetallic regions. According to this classification, both Area C and Area D at the amorphous/Ni60 interface were mixtures of the amorphous and intermetallic phase, and the closer it was to the center of Ni60, the more intermetallic compounds there were. More intermetallic compound structures were detected at the amorphous/WC interface by the calibration of SAED, such as FeNi_3_, Ni_4_B_3_ and Cr_3_C_2_, which matched the ultra-high nanoindentation hardness. The same hardening behavior was also found in [21], where the cross-sectional hardness of the coating increased from 660 to 870 HV after adding 8% WC/12Co.

As a completely elastic-perfectly plastic solid material, Fe-based amorphous coating follows the plastic volume conserving characteristic, and so the indentation was deeper [29,30], while the interface area with shallow indentation obviously had a smaller plastic deformation due to the hardness and brittleness of the intermetallic compound. Intermetallic compounds exhibit higher hardness, higher elastic moduli and lower Poisson ratios than amorphous plastic structures [31], so the indentation in Figure 4c was smaller both in depth and springback depth. Similar precipitation of nanocrystals of intermetallic compounds was also reported during the cooling process of NiCrBSi coating [32]. Chromium carbide is the first to precipitate due to its high melting point, and therefore, Ni is enriched in the residual liquid metal, resulting in the tendency of FeNi_3_ to form between carbide grain boundaries. Studies [13] have shown that these fine nanocrystals dispersed in the alloy layer and improved the strength of the alloy by inhibiting the dislocation motion. Although there are no grain boundaries and dislocations in the macroscopic amorphous structure, these fine grains exist in the limited crystal areas around the Ni transition layer, resulting in lattice distortion through the mutual dissolution of Fe and Ni atoms, thus increasing the mechanical properties. Therefore, these undesired structural inhomogeneities in the amorphous structure did not deteriorate local properties but also had high strength, thus maintaining the favorable bonding of the amorphous/WC interface.

According to the wear test and the previous test results of authors [33], the process of amorphous coating wear is depicted as a schematic in Figure 9. At the initial stage, running-in wear occurred when the friction ball contacts the coating surface and flattens the protruding particles, as shown in Figure 9b.

If the melting of sprayed particles is not sufficient [34] or the porosity is high [35], there will be protruding semi-fused particles or pores on the coating surface, which will exert resistance to the friction pair. Therefore, the COF often fluctuates violently in the running-in stage. The large fragments sheared by the friction pair followed the ZrO_2_ ball and carved furrows on the coating surface, as shown in Figure 6a, indicate that the running-in stage was accompanied by abrasive wear. Such abrasive wear caused by peeling particles of the coating has been reported in [17]. Particles that have poor bonding with the pre-deposited particles warped under periodic load aggravated the generation of abrasive particles during fatigue wear. However, because WC was anchored deeper into the coating-like nails, the spraying particles close to WC were less warped, and the abrasive particles are gradually entrapped when passing through WC particles, which are ground to fine debris under frequent loads. In the middle wear stage, the WC and ZrO_2_ surface also began to break into fragments, and the coating surface started to undergo oxidation wear due to the accumulation of friction heat. Coated abrasive debris, ZrO_2_ fragments and oxidation products in the amorphous coating were embedded in the dents from the fatigue wear from Figure 6b, but this debris, along with WC fragments, in the composite coating were also embedded in the diffused Ni layer, as shown in Figure 6i. The diffused Ni transition trapped fragments and debris, and it adapted to periodic loads by plastic deformation, as shown in Figure 6f. Under this mechanism of the reinforcing phase, the fatigue wear and abrasive wear of the composite coating surface were much less serious than that of the amorphous coating, of which the COF changes dramatically due to the serious abrasive wear and frequent fatigue failure in the middle wear stage. As a result, the composite coating maintains a flat surface and a more stable COF than the amorphous coating. Similarly, the addition of 20% ZrO_2_ by a mass fraction can effectively reduce the wear of ZrO_2_-reinforced Fe-based amorphous composite coatings [17].

As shown in Figure 8, although the corrosion resistance of the composite coating was stronger than SS304 stainless steel, it was deteriorated to some extent due to the addition of reinforcing phase particles.

According to the relation below [36], the corrosion rate (C_R_) is given by:C_R_ = I_corr_ × A_w_ × 10 × 3.15 × 10^7^/(n × F × d)(1)
where, A_w_ is the metal atomic weight, n is the number of charge transfer when metal corrodes, F is the Faraday’s constant and d is the density of the alloy, the C_R_ has a positive correlation with the I_corr_, so it could be inferred that the charge transfer of the composite coating was more active than that of the amorphous coating. The corrosion resistance of amorphous coatings is mainly based on microstructure uniformity and composition homogeneity. Therefore, the addition of Ni-WC inevitably led to inhomogeneity on the coating surface; thus, providing the prerequisite for the formation of galvanic cells. Fortunately, the E_corr_ of the composite coating did not significantly decrease, indicating that Ni-WC particles did not change the overall corrosion tendency. The reason why the corrosion resistance of the composite coating did not decrease seriously may be that the filling effect of the reinforcing phase particles reduced the porosity to less than 1.22%, which has been reported as a key threshold for the Fe-based amorphous coating. Above this value, the corrosion resistance is more dominated by porosity, while below this value, it is dominated by the amorphous content. As mentioned above, if the second phase particles in the composite coating have an appropriate size, which is slightly smaller than that of the amorphous particles, they can fill the cavities and pores caused by the insufficient plastic deformation between the amorphous particles [18], and thus, make the porosity low enough to reduce the probability of the Cl^-^ enrichment. The addition of Al_2_O_3_ has also been reported to reduce the infiltration of Cl^-^ through the pores and suppressed corrosion of the coating in the NaCl solution [37]. However, in the coating with Al_2_O_3_-TiO_2_ particles, the corrosion resistance is not as good as stainless steel in an NaOH solution [18]. Combined with the overall performance of the Tafel polarization plots in HCl solution, the composite coating has better corrosion resistance than SS304 and WE43, although its performance is not as good as that of a completely amorphous coating based on the testing methods of this study.

## 5. Conclusions

In this study, an Fe-based amorphous coating and an Fe-based amorphous composite coating were successfully prepared on the surface of a magnesium alloy by high-velocity oxygen-fuel spraying. Through the wear test, Tafel polarization plots and microscopic interface bonding mechanism analysis, the following conclusions were reached:The amorphous content of the composite coating exceeds 90%, and the porosity is approximately 1%. Compared with the completely amorphous coating, the composite coating is more dense and only slightly crystallized.The COF of the composite coating was 0.3, which is less than the 0.37 associated with the amorphous coating and only half of that of the WE43 magnesium alloy substrate. The Ni transition layer enhanced the bonding between the WC reinforcing phase particles and amorphous coating during wear and thus improved the wear resistance.The E_corr_ and I_corr_ of the composite coating were −0.331 V and 3.582 μA, respectively, and the addition of the reinforcing phase particles did not significantly deteriorate the corrosion tendency and corrosion rate, which was still better than that of SS304 stainless steel and WE43 magnesium alloy.The amorphous/Ni60 interface and the amorphous/WC interface in the composite coating have achieved favorable metallurgical bonding, where Fe and Ni were fully diffused. A mixture of amorphous and nanocrystals existed in the Ni transition layer, and the nanocrystals did not deteriorate the bonding properties but exhibited excellent mechanical properties.

## Figures and Tables

**Figure 1 materials-14-06571-f001:**
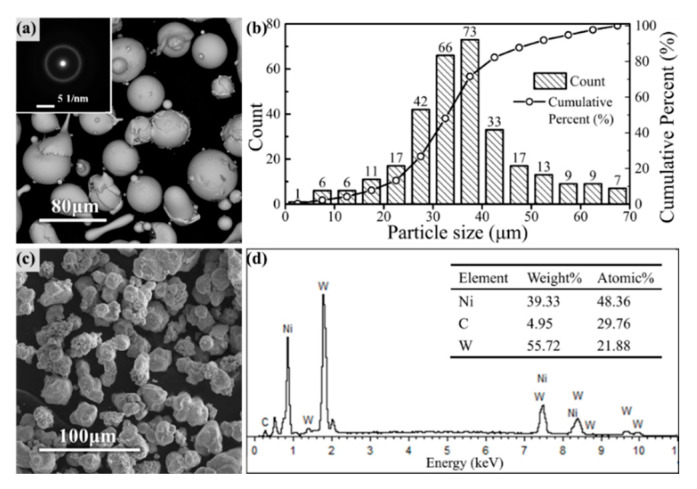
SEM micromorphology of the feedstock powders: (**a**) surface morphology and the SAED pattern of Fe-based amorphous powders; (**b**) granulometric distribution of Fe-based amorphous powders; (**c**) surface morphology of the reinforcing phase particles; (**d**) chemical component of the WC wrapped by Ni.

**Figure 2 materials-14-06571-f002:**
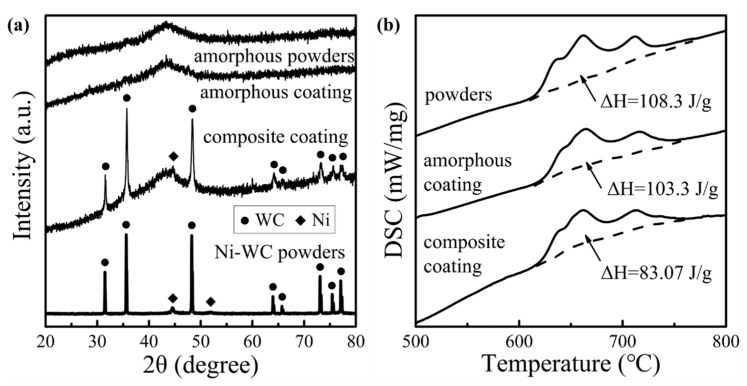
The phase characteristics and thermodynamic analysis of amorphous particles and HVOF coatings: (**a**) the XRD patterns; (**b**) the DSC curves.

**Figure 3 materials-14-06571-f003:**
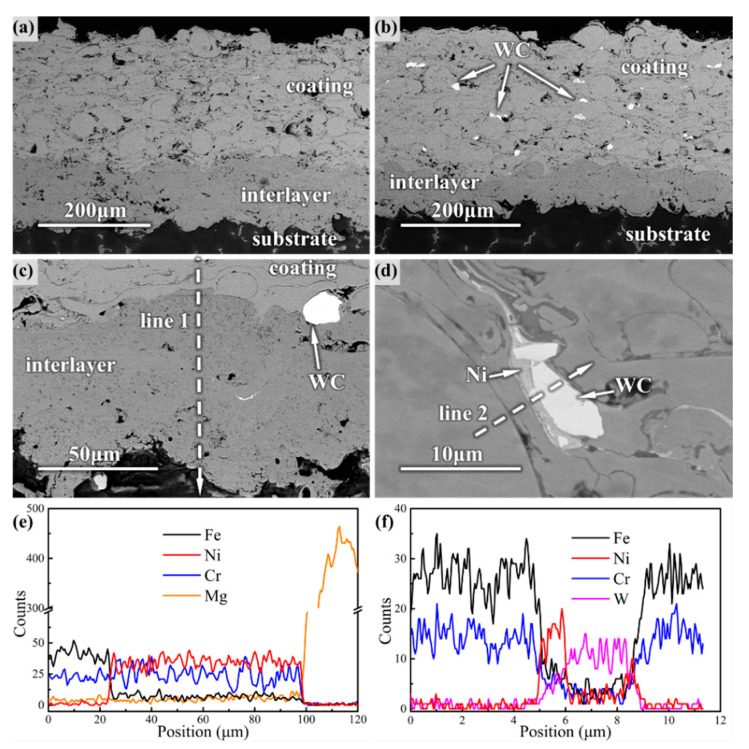
SEM morphology and EDS analysis of the coating cross-section: (**a**) SEM micromorphology of the amorphous coating; (**b**) SEM micromorphology of the composite coating; (**c**,**e**) interlayer elemental line scanning in the direction of line 1; (**d**,**f**) reinforcing phase particle elemental line scanning in the direction of line 2.

**Figure 4 materials-14-06571-f004:**
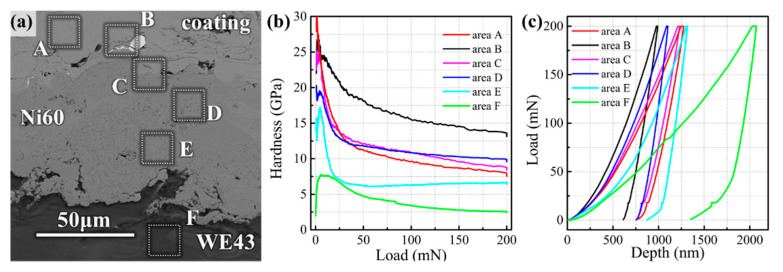
Mechanical properties at different areas of the coating cross-section: (**a**) selected areas of nanoindentation; (**b**) hardness-load curves; (**c**) load-depth curves.

**Figure 5 materials-14-06571-f005:**
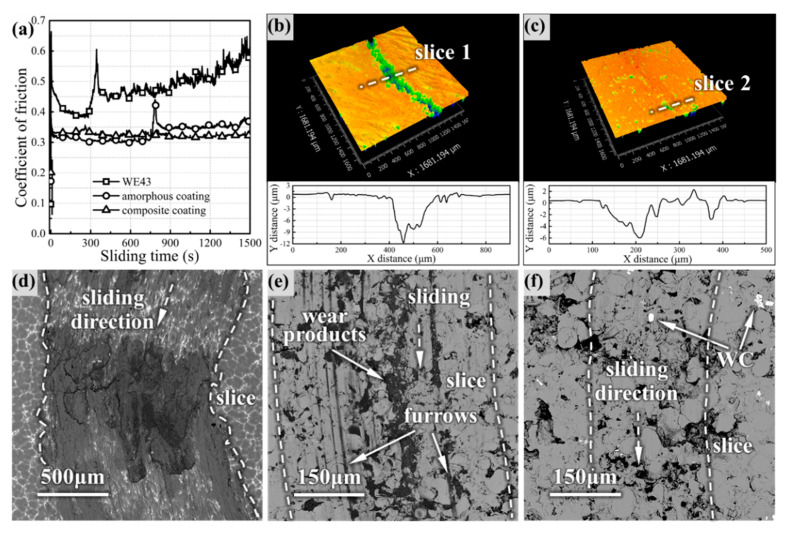
Wear test of WE43 and coatings: (**a**) COF of WE43 and Fe-based amorphous coatings; (**b**) and (**c**) 3D worn surface topography of the amorphous coating and composite coating, respectively; (**d**–**f**) worn surface SEM morphology of WE43 and coating 2.

**Figure 6 materials-14-06571-f006:**
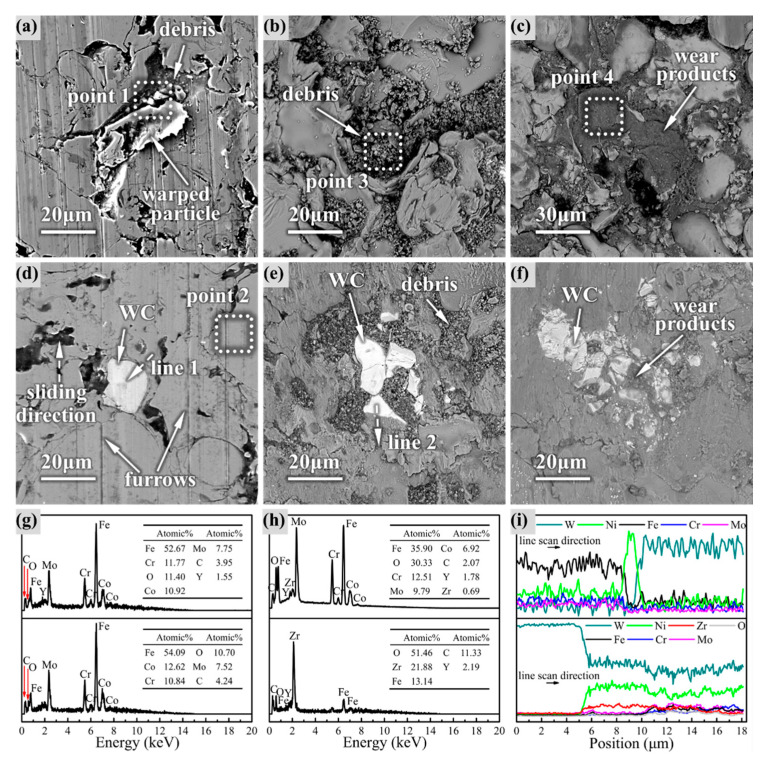
Worn morphology and EDS analysis of the coating surface after different wear times: (**a**–**c**) amorphous coating after 5, 15, 25 min of wear; (**d**–**f**) composite coating after 5, 15, 25 min of wear; (**g**) EDS after 5 min of wear at point 1 and point 2; (**h**) debris EDS after 10 min of wear at point 3, and wear products EDS after 15 min of wear at point 4; (**i**) elemental line scanning in the direction of line 1 after 5 min of wear and in the direction of line 2 after 10 min of wear.

**Figure 7 materials-14-06571-f007:**
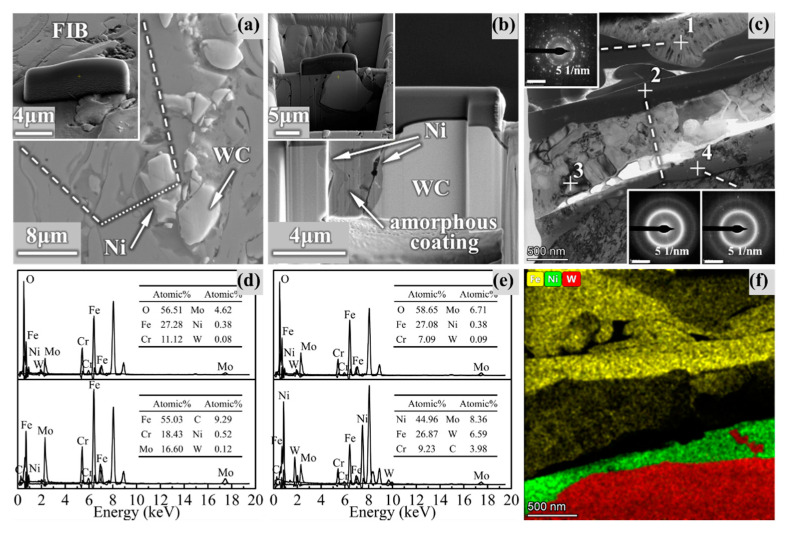
Microstructure analysis of the interface of amorphous/reinforcing phase particles: (**a**,**b**) the process of selecting target region and preparing TEM sample by FIB thinning technique; (**c**) morphology and SAED pattern of the Ni transition layer at the amorphous/WC interface; (**d**) EDS at point 1 and point 2; (**e**) EDS at point 3 and point 4; (**f**) map scanning of the main elements around the interface.

**Figure 8 materials-14-06571-f008:**
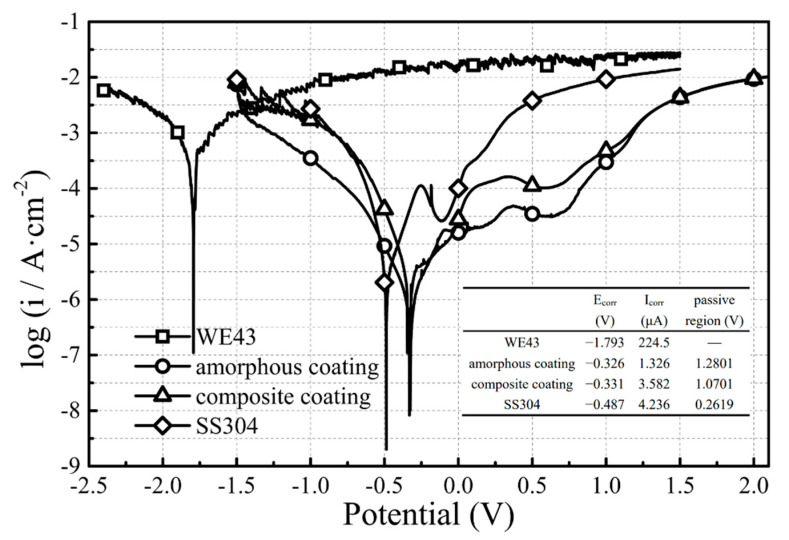
Tafel polarization plots of WE43, SS304 and amorphous coatings in 0.1 mol/L HCl solutions.

**Figure 9 materials-14-06571-f009:**
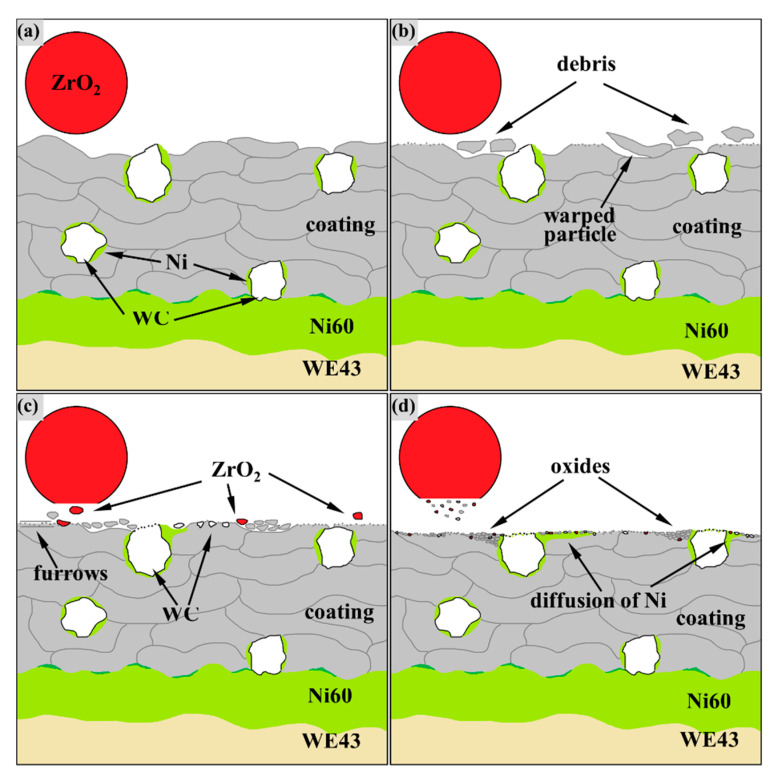
The schematic of amorphous coating wear process: (**a**) the contact of the coating and the ZrO^2^ friction ball; (**b**) fragments from sheared protruding particles and warped particles in the running-in stage; (**c**) the fragments are ground into debris, and the Ni layer begins to spread in the middle wear stage; (**d**) the products of oxidation wear are well mixed with the Ni layer after diffusion.

**Table 1 materials-14-06571-t001:** Chemical compositions of the feedstock powders in HVOF (mass fraction/%).

Powders	Fe	Ni	Cr	Co	Mo	Cu	C	B	Si	Y
Amorphous Powders	surplus	-	14.95	8.57	26.9	-	3.2	1.28	-	3.01
Ni60	≤5	surplus	17.0–20.0	-	2.0–3.0	2.0–3.0	0.7–1.1	3.0–4.0	3.5–5.0	-

**Table 2 materials-14-06571-t002:** Spraying parameters of the HVOF process.

Powders	Oxygen Flow Rate (m^3^/h)	Fuel Flow Rate (L/h)	Feed Rate (g/min)	Spray Distance (mm)	Spray Gun Size (mm)	Spraying Rate (m/min)
Amorphous Coatings	46	14	35	320	127.0	20
Ni60 interlayer	52	24	60	320	127.0	30

**Table 3 materials-14-06571-t003:** Mechanical properties at different areas of the coating cross-section measured by the nanoindentation test.

Area	Hardness (GPa)	Modulus (GPa)	Depth (nm)
A	8.81	139.4	1275
B	14.6	199.5	989
C	9.81	140.5	1231
D	10.37	203.3	1102
E	6.48	150.7	1316
F	2.9	80.9	2070

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
