# Peer review of "Reinforcing Mechanism of WC Particles in Fe-Based Amorphous Matrix Coating on Magnesium Alloy Surface"

_materials, 2021, doi:10.3390/ma14216571_

Round 1
Reviewer 1 Report
Review for
materials-1407400
Reinforcing mechanism of WC particles in Fe-based amorphous matrix coating on magnesium alloy surface
The coating study on Mg is still a very significant topic. This article has abundant research data, good surface characterization, and in-detail analysis.
I recommend it for publishing after minor revision. I think, nonetheless, that the manuscript could be improved if the authors could address the comments and recommendations I listed below.
The novelty of this research should be highlighted in the Abstract.
Line 40: Mg does not have good corrosion resistance owing to the low P-B
ratio that can not form a protective passive film. This is the most significant reason and you should mention it in your article.
I assume your research is focusing on the application of aerospace. You should add some background information about Mg in the aerospace industry.
Cause you introduced a lot of elements in your coating, how's the micro-galvanic corrosion? Will the micro-galvanic corrosion deteriorate your coating system?
You only rely on polarization curves to describe its corrosion resistance performance, it is not enough to track its long-time performance. At least, you should add an EIS test to compare the corrosion-resistant changes. Additionally, if your Mg materials going to be used in aircraft, you may need salt spray tests.
Figure 8. Why you tested your sample in 0.1M HCl solution?
Line 433. You believe your sample has a better corrosion resistance than SS304, but in my point of view, after a long time of active service (, especially for the potential galvanic corrosion, attach or coating defect),it may have some localized corrosion. Please revise your description, you may say that based on your testing methods/environment is better than SS304.
Author Response
Response to Reviewer 1 Comments
Reinforcing mechanism of WC particles in Fe-based amorphous matrix coating on magnesium alloy surface
The coating study on Mg is still a very significant topic. This article has abundant research data, good surface characterization, and in-detail analysis.
I recommend it for publishing after minor revision. I think, nonetheless, that the manuscript could be improved if the authors could address the comments and recommendations I listed below.
Point 1: The novelty of this research should be highlighted in the Abstract.
Response 1: Thanks to the reviewer's suggestion, the highlight of this study is mainly the preparation of dense Fe-based amorphous composite coating with high amorphous content (> 90%) on the magnesium alloy surface. The Ni transition layer is proved to exist in the form of amorphous/nanocrystalline mixture structure. This structure has been proved to have good mechanical properties by previous studies, so it provides support for the wear process schematic proposed in this study. Besides, due to the filling effect of fine reinforcing particles and the partly amorphous structure of Ni transition layer, the addition of reinforcing particles does not seriously deteriorate the corrosion resistance of the coating. The authors have added some necessary data in the abstract to demonstrate the novelty of this study more clearly. The modified content is in line 13 and 21, as shown below:
The porosity was ~1% and the amorphous content exceeded 90%.
Reinforcing particles were confirmed to form metallurgical bonding with the coating.
Point 2: Line 40: Mg does not have good corrosion resistance owing to the low P-B ratio that can not form a protective passive film. This is the most significant reason and you should mention it in your article.
Response 2: Thanks to reviewer for the professional comment. Indeed, as the reviewer said, Mg is difficult to form a dense passivation film due to its low pilling-bedworth ratio. After careful consideration, we believe that the low P-B ratio is indeed a factor that cannot be ignored for the poor corrosion resistance of Mg. Thus, we have made a supplement in the introduce section. The modified content is in line 43, as shown below:
However, magnesium alloy is bound to have poor wear resistance, and can not form a dense passivation film against the corrosion owing to its softness, active chemical properties [3] and low pilling-bedworth ratio [4]. And hence researchers have to enhance its surface properties by alloying or surface protection [5].
Point 3: I assume your research is focusing on the application of aerospace. You should add some background information about Mg in the aerospace industry.
Response 3: Thanks to the reviewer for the comment. The aerospace field is indeed a promising field for magnesium alloy application, and we have added the literature related to magnesium alloy application. The modified content is in line 41, as shown below:
Replacing aluminum aircraft seats with magnesium, for example, can reduce body weight by 28-30%[2].
Point 4: Cause you introduced a lot of elements in your coating, how's the micro-galvanic corrosion? Will the micro-galvanic corrosion deteriorate your coating system?
Response 4: Thanks to the reviewer for the comment. In the Fe-based amorphous system, Co, Cr, Mo and other elements are added in order to increase the formation of the glass forming ability (GFA). The amorphous system has microstructural uniformity and compositional homogeneity, so the passivation elements such as Cr and Mo are able to form a dense and uniform passivation film. Thus, the surface potential of amorphous coating is uniform, and the micro-galvanic corrosion is not serious. The key factor in this study is the addition of reinforcing phase particles. WC and the Ni layer have different potentials different from the coating, which theoretically tend to form galvanic cells. Fortunately, the corrosion resistance of coating did not deteriorate, because of the filling effect of fine reinforcing particles and the mixture structure of amorphous/nanocrystals around the Ni transition layer, which showed favorable properties. Therefore, it is fundamentally acceptable that the composite coating has the same corrosion behavior as amorphous coating, and the micro-galvanic corrosion did not deteriorate the corrosion resistance.
Point 5: You only rely on polarization curves to describe its corrosion resistance performance, it is not enough to track its long-time performance. At least, you should add an EIS test to compare the corrosion-resistant changes. Additionally, if your Mg materials going to be used in aircraft, you may need salt spray tests.
Response 5: Thanks to the reviewer for the meticulous comment. As the reviewer pointed out, electrochemical corrosion is a complex process. Tafel curve is a typical characterization. At present, the main function of reinforcing phase particles is to enhance the wear resistance of coatings. Meanwhile, the corrosion behavior of Fe-based amorphous coating, especially the passivation and Cl- erosion behavior, has many similarities with stainless steel, so this study chose to simulate this behavior in HCl solution. In this experiment, Tafel plots can show that the addition of reinforcing particles does not lead to a serious deterioration of the corrosion resistance of the coating. Therefore, for the protection of magnesium alloy surface, the composite coating in this study has achieved the expected effect. We are also actively conducting further research on the effect of particle relative corrosion resistance.
Point 6: Figure 8. Why you tested your sample in 0.1M HCl solution?
Response 6: Thanks to reviewer for the comment. HCl solution is a relatively stable environment for chloride ion corrosion due to the similar composition elements and passivation behavior of Fe-based amorphous coatings and stainless steel. However, 0.1mol/L HCl is close to the actual application environment, and the chloride ion concentration is also suitable to simulate the enrichment effect of chloride ion stably. Therefore, the suitable test condition was set as 0.1mol/L.
Point 7: Line 433. You believe your sample has a better corrosion resistance than SS304, but in my point of view, after a long time of active service (, especially for the potential galvanic corrosion, attach or coating defect),it may have some localized corrosion. Please revise your description, you may say that based on your testing methods/environment is better than SS304.

Response 7: Thanks to the reviewer for the rigorous comment. The comparison with stainless steel should be established under specific conditions. Therefore, after our thinking and discussion, it is necessary to add that the corrosion resistance of coating is better than stainless steel under the test method of this experiment. The modified content is in line 444, as shown below:
Combined with the overall performance of the Tafel polarization plots in HCl solution, the composite coating has better corrosion resistance than SS304 and WE43, although its performance is not as good as that of a completely amorphous coating based on the testing methods of this study.
Thanks again to the reviewer and editors for the support of this study. Wish everything goes well with your work!

Reviewer 2 Report
The authors report the microstructure, hardness, corrosion resistance, and friction coefficient of Fe+WC+Ni composite coating on the magnesium alloy surface. The Ni layer plays a role in bonding between Mg and Fe. The composite coating is stable, and the addition of WC does not deteriorate the corrosion resistance.
The coating of Mg is an important research area. The composite coating investigated would be original and shows good performance, which is valuable for the publication. I recommend the minor revision. Please address the minor issues listed below.
- Please compare the performance of corrosion resistance, hardness, and friction coefficient between the present coating and the other coating with the reinforcing particles such as Al2O3, ZrO2, and B4C on Mg.
- Section title 3. Result and 4. Discuss might be 3. Results and 4. Discussion, respectively.
Author Response
Response to Reviewer 2 Comments
The authors report the microstructure, hardness, corrosion resistance, and friction coefficient of Fe+WC+Ni composite coating on the magnesium alloy surface. The Ni layer plays a role in bonding between Mg and Fe. The composite coating is stable, and the addition of WC does not deteriorate the corrosion resistance.
The coating of Mg is an important research area. The composite coating investigated would be original and shows good performance, which is valuable for the publication. I recommend the minor revision. Please address the minor issues listed below.
Point 1: Please compare the performance of corrosion resistance, hardness, and friction coefficient between the present coating and the other coating with the reinforcing particles such as Al2O3, ZrO2, and B4C on Mg. 

Response 1: Thanks to reviewer for the thoughtful consideration. Based on the careful understanding of our authors, we think it is necessary to add the studies with similar experimental conditions into the discussion of this study as a comparison. We added some new literatures to compare the coating performance with the addition of different reinforcing phase particles, and relevant supplements have been marked in the paper respectively. The modified content is in line 363, 399, 417 and 440, as shown below:
The same hardening behavior was also found in the study [21], where the cross-sectional hardness of the coating increased from 660 HV to 870 HV after adding 8% WC/12Co.
Such abrasive wear caused by peeling particles of the coating has been reported in the study [17]
Similarly, the addition of 20 % ZrO2 by mass fraction can effectively reduce the wear of ZrO2 reinforced Fe-based amorphous composite coatings [17].
As mentioned above, if the second phase particles in the composite coating have an appropriate size that is slightly smaller than that of the amorphous particles, they can fill the cavities and pores caused by the insufficient plastic deformation between the amorphous particles [16], thus make the porosity low enough to reduce the probability of the Cl- enrichment. The addition of Al2O3 has also been reported to reduce the infil-tration of Cl- through the pores and suppressed corrosion of the coating in NaCl solu-tion [37]. But in the coating with Al2O3-TiO2 particles, the corrosion resistance is not as good as stainless steel in NaOH solution [18].
Point 2: Section title 3. Result and 4. Discuss might be 3. Results and 4. Discussion, respectively.
Response 2: Thanks to reviewer for reminding us. The authors apologize for the inappropriateness of the title. And the title has been revised accordingly.
Thanks again to the reviewer and editors for the support of this study. Wish everything goes well with your work!
